# The Wdr5-H3K4me3 Epigenetic Axis Regulates Pancreatic Tumor Immunogenicity and Immune Suppression

**DOI:** 10.3390/ijms25168773

**Published:** 2024-08-12

**Authors:** Kaidi Deng, Liyan Liang, Yingcui Yang, Yanmin Wu, Yan Li, Rongrong Zhang, Yulin Tian, Chunwan Lu

**Affiliations:** 1School of Life Sciences, Tianjin University, Tianjin 300072, China; dengkd991226@163.com (K.D.); lly_lly@tju.edu.cn (L.L.); 2021226047@tju.edu.cn (Y.Y.); wym971220@163.com (Y.W.); yanli_@tju.edu.cn (Y.L.); 2Institute of Materia Medica, Peking Union Medical College, Beijing 100050, China; zrongrong@imm.ac.cn (R.Z.); tianyulin@imm.ac.cn (Y.T.)

**Keywords:** WDR5, MLL1, H3K4me3, MHC I, HLA, immunogenicity, immune suppression, pancreatic tumor

## Abstract

The WDR5/MLL1-H3K4me3 epigenetic axis is often activated in both tumor cells and tumor-infiltrating immune cells to drive various cellular responses in the tumor microenvironment and has been extensively studied in hematopoietic cancer, but its respective functions in tumor cells and immune cells in the context of tumor growth regulation of solid tumor is still incompletely understood. We report here that WDR5 exhibits a higher expression level in human pancreatic tumor tissues compared with adjacent normal pancreas. Moreover, WDR5 expression is negatively correlated with patients’ response to chemotherapy or immunotherapy in human colon cancer and melanoma. However, WDR5 expression is positively correlated with the HLA level in human cancer cells, and H3K4me3 enrichment is observed at the promoter region of the HLA-A, HLA-B, and HLA-C genes in pancreatic cancer cells. Using mouse tumor cell lines and in vivo tumor models, we determined that WDR5 deficiency or inhibition significantly represses MHC I expression in vitro and in vivo in pancreatic tumor cells. Mechanistically, we determine that WDR5 deficiency inhibits H3K4me3 deposition at the MHC I (H2K) promoter region to repress MHC I (H2K) transcription. On the other hand, WDR5 depletion leads to the effective downregulation of immune checkpoints and immunosuppressive cytokines, including TGFβ and IL6, in the pancreatic tumor microenvironments. Our data determine that WDR5 not only regulates tumor cell immunogenicity to suppress tumor growth but also activates immune suppressive pathways to promote tumor immune evasion. Selective activation of the WDR5-MHC I pathway and/or selective inhibition of the WDR5–immune checkpoint and WDR5–cytokine pathways should be considered in WDR5-based epigenetic cancer immunotherapy.

## 1. Introduction

The dysregulation of epigenetic mechanisms, including DNA methylation and histone modification, can lead to various downstream pathogenesis and recently has been recognized as one of the novel hallmarks of cancer progression [1,2]. Among all the epigenetic modifications, histone methylation is the most well-studied histone modification pattern [3]. H3K4, H3K9, H3K27, H3K36, H3K79, and H4K20 are essential methylated sites that can be modified into mono-, di-, and tri-methylation states [4], and the variety of histone methylations constitutes a complicated regulatory network to engage in cancer development [5]. Especially, H3K4 trimethylation is a near-universal histone modification in tumorigenesis [6,7,8]. H3K4me3 has been defined as a transcription-activated mark, usually depositing at promoter regions and transcription start sites of target genes [9]. Emerging data have determined that lysine methyltransferases (KMTs), which specifically catalyze H3K4 methylation, are extensively involved in cancer immunity regulation [10,11].

MLL1 (Mixed-Lineage Leukemia 1, also called MLL, KMT2A, HRX, HTRX, and ALL1) complex is one of the KMTs responsible for catalyzing H3K4 trimethylation [12]. The constitutive activation of the MLL1-H3K4me3 pathway can directly lead to human acute lymphoid leukemia (ALL) and acute myeloid leukemia (AML) [13,14]. Nevertheless, the catalytic activity of MLL1 alone is super low. To achieve the optimal histone methyltransferase (HMTase) activity, MLL1 usually forms a complex with other proteins, such as WDR5, Ash2L, RBP5, and DPY30 [15]. WDR5 is a key non-catalytic component in MLL1 complex, which is indispensable for the execution of KMTs activity of MLL1 [16,17]. Therefore, targeting the interaction between WDR5 and MLL1 to inhibit the MLL1 activity of catalyzing H3K4me3 is a potential effective therapeutic strategy for ALL and AML [16,17]. Recent studies demonstrated that targeting the interaction between WDR5 and MLL1 via the WDR5-specific inhibitor WDR5-47 can effectively antagonize MLL1 activity of catalyzing H3K4me3 enrichment, resulting in the downregulation of target genes such as PD-L1 and OPN to activate CTLs’ function and suppress pancreatic tumor growth [18,19]. These findings thus extended the regulatory role of WDR5/MLL1-H3K4me3 axis from leukemia to solid tumor.

WDR5-H3K4me3 has a wide variety of regulatory roles in the cancer-immunity cycle and has been recognized as a cellular multi-masker [20,21]. H3K4me3 not only modulates anti-cancer immunity, such as enhancing cancer-antigen [22,23] and MHC molecules’ expression [24,25], promoting T-cell development [26,27,28,29] and chemokine-induced T-cell trafficking [30], improving DCs’ maturation [31], but also regulates pro-cancer immunity, such as promoting immune-checkpoint expression [18,19,32,33], supporting M2 polarization of TAMs [34,35], and facilitating immunosuppressive functions of MDSCs [36,37,38]. Although the function of the WDR5/MLL1-H3K4me3 axis has been extended from leukemia to solid tumors such as pancreatic cancer [18,19], the relative functions of the WDR5/MLL1-H3K4me3 axis in tumor cells and immune cells in solid tumor is incompletely studied.

We report here that WDR5 has a higher expression level in human pancreatic tumor tissues, and WDR5 expression is negatively correlated with cancer patients’ response to chemotherapy/immunotherapy in human colon cancer and melanoma. Unexpectedly, WDR5 expression exhibits a positive correlation with HLA level in breast-cancer patients, and notable H3K4me3 deposition is observed at the promoter region of HLA-A, -B, and -C. Furthermore, Wdr5 deficiency/inhibition represses MHC I expression in vitro and in vivo. Mechanistically, we determined that Wdr5 deficiency restrains H3K4me3 binding to the MHC I promoter region to suppress MHC I transcription. On the other hand, Wdr5 depletion leads to the downregulation of immune checkpoints, including PD-1, PD-L1, and Spp1; and immunosuppressive cytokines, including TGFβ and IL6, in the tumor microenvironments to overcome the pro-cancer effect caused by Wdr5 deficiency-induced MHC I transcription inhibition. These findings suggested that selective activation of the WDR5-MHC I pathway and/or selective inhibition of the WDR5–immune checkpoint/cytokine pathways should be taken into consideration in WDR5-based epigenetic cancer immunotherapy.

## 2. Results

### 2.1. WDR5 Expression Profiles in Human Cancer

To determine the human relevance of this study, we first analyzed the TCGA database for WDR5 expression in the RNA level of different human tumor types. An analysis of the dataset revealed that WDR5 exhibits higher RNA level in pancreatic tumor tissues than in adjacent normal pancreas (Figure 1A). Due to the unavailability of human pancreatic tumor scRNA-seq datasets, we then mined scRNA-seq datasets (GSE178341) [39] and analyzed WDR5 expression profiles in human colon tumors. Colon tumor resident cells are annotated in Figure 1B. The cellular subtype analysis demonstrated that WDR5 is widely expressed in almost all cell subpopulations, while expression in epithelial and plasma cells is relatively lower than expression in other cell subsets (Figure 1B). The similar phenomenon can be found in human breast-cancer scRNA-seq datasets (GSE176078) [40] (Appendix A).

To elucidate the functional link between WDR5 expression and response to chemotherapy, we next analyzed datasets of a recently published human colon cancer-patient clinical trial for combined PD-1, BRAF, and MEK inhibition [41]. A remarkably elevated WDR5 expression was observed in non-responders as compared to responders (Figure 1C). We further mined datasets of another published human melanoma-patient clinical trial for checkpoint immunotherapy (GSE120575) [42]. Similarly, a significantly increased WDR5 expression was observed in non-responders as compared to responders (Appendix A).

### 2.2. WDR5 Expression Positively Correlates with HLA Expression in Human Cancer

HLA-I, like the MHC I in mouse, regulates the immunogenicity of human cancer cells [43]. To verify WDR5 expression in human pancreatic cancer patients, we analyzed WDR5 expression in five pairs of human pancreatic tumor tissues and adjacent normal pancreas. Both the IHC (Figure 2A) and PCR (Figure 2B) analyses showed that WDR5 exhibits a significantly higher expression in human pancreatic tumor tissues compared with adjacent normal pancreas. An available human breast-cancer scRNA-Seq dataset (GSE176078) [40] was then used to further determine the link between WDR5 and HLA at the single-cell level. Interestingly, our correlation analysis identified a significantly positive correlation between the WDR5 expression level and the expression levels of HLA-A, HLA-B, and HLA-C (Figure 2C).

It is well documented that WDR5 is an essential non-catalytic component of MLL1 complex that helps MLL1 achieve optimal activity when catalyzing H3K4 trimethylation [16,17]. We then mined the Encode dataset (https://www.encodeproject.org, accessed on 4 April 2024) and analyzed H3K4me3 enrichment at the HLA-A, -B, and -C promoter regions in human pancreatic tumor cell line PANC1. As expected, H3K4me3 was deposited at the promoter regions of HLA-A, -B, and -C in PANC1 cells (Figure 2D). In addition, the H3K4me3 is mostly enriched at the downstream of the transcription start site of HLA-A, -B, and -C (Figure 2D).

### 2.3. Wdr5 Regulates MHC I Expression in Pancreatic Tumor Cells In Vitro

To validate the above findings in mouse cells, we made use of mouse pancreatic tumor cell line PANC02. To determine whether Wdr5 directly regulates MHC I expression, Wdr5 was knocked out in PANC02 cells, using the CRISPR technique. PCR and Western blotting verified that Wdr5 has been efficiently knocked out both in PANC02-gRNA1 and PANC02-gRNA2 cells compared with PANC02-Scramble cells, while PANC02-gRNA2 cells have higher efficiency of knocking out by dramatically suppressing Wdr5 and H3K4me3 expression (Figure 3A). Therefore, we chose PANC02-gRNA2 cells for further functional analysis. As expected, depleting Wdr5 significantly abolished H2Kb expression in both RNA level (Figure 3B) and protein level (Figure 3D). However, Wdr5 deficiency in tumor cells has minimal effects on tumor cell growth in vitro (Figure 3C).

### 2.4. Wdr5 Controls MHC I Expression in Pancreatic Tumor Cells In Vivo

To determine whether the above results can be translated to pancreatic tumor growth control and MHC I expression regulation in vivo, we subcutaneously injected PANC02-Scramble and PANC02-gRNA2 tumor cells into syngeneic C57BL/6 mice to establish an in vivo pancreatic tumor model. An analysis of H3K4me3 showed that the ablation of Wdr5 caused attenuated H3K4me3 expression in total tumor tissues (Figure 3F). Unexpectedly, the depletion of Wdr5 did not change tumor size and weight dramatically (Figure 3E). To further demonstrate the regulatory function of Wdr5 on MHC I molecules in vivo, we analyzed the MHC I expression in total tumor tissues. The depletion of Wdr5 dramatically suppressed MHC I (H2Kb/H2Db) expression in the RNA level in the total tumor tissues (Figure 3G). We also isolated CD45^+^ and CD45^−^ cells from total tumor tissues with positive selection beads. In CD45^+^ cells, both CD4 and CD8 expression increased after Wdr5 knockout in RNA level, especially CD4 (Figure 3H). In CD45^−^ cells, both H2Kb and H2Db downregulated significantly after deletion of Wdr5 (Figure 3I).

To validate the regulation of MHC I by Wdr5 in protein levels, we analyzed H2Kb/H2Db expression in CD45^−^ cells from total PANC02 tumor tissues by flow cytometry. As expected, the ablation of Wdr5 significantly downregulated H2Kb/H2Db expression in protein level (Figure 4A). Nevertheless, CD8^+^ (Figure 4B) and CD4^+^ (Figure 4C) cells’ tumor infiltration was enhanced dramatically after the depletion of Wdr5, which is consistent with elevated CD8/CD4 expression in RNA level (Figure 3H).

### 2.5. Inhibition of Wdr5 Represses MHC I Expression

To strengthen the above findings, we took a complimentary approach by culturing PANC02 cells in the presence of WDR5 specific inhibitor WDR5-47. It is reported that WDR5-47 can block the interaction between WDR5 and MLL1 to antagonize enzyme activity of MLL1 to catalyze H3K4 trimethylation, leading to the consequent suppressive expression of target genes regulated by the WDR5/MLL1-H3K4me3 axis [18]. As expected, WDR5-47 decreased the H3K4me3 level dose dependently in PANC02 cells (Figure 5A) and inhibited the tumor cell viability both dose dependently and time dependently (Figure 5B). WDR5-47 also suppressed H2Kb/H2Db expression in vitro dramatically at 100 μM (Figure 5C). For the mouse pancreatic tumor model in vivo, the inhibition of Wdr5 significantly repressed pancreatic tumor growth in tumor weight (Figure 5D) and effectively downregulated the H3K4me3 level in total tumor tissues (Figure 5E).

### 2.6. Wdr5-H3K4m3 Axis Regulates MHC I Expression in Mouse Pancreatic Tumor Cells

In accordance with our above observations, WDR5 expression is positively correlated with HLA expression (Figure 2C), and H3K4me3 enriched in HLA promoter regions (Figure 2D) suggested that Wdr5 may regulate MHC I expression by controlling H3K4me3 enrichment at promoter regions of MHC I. To test this hypothesis, we first mined the Encode database and analyzed H3K4me3 enrichment at the MHC I promoter region. H3K4me3 deposition was observed at the H2K promoter region in the mouse melanoma MEL cells (Figure 6A). Then, we performed a ChIP analysis of PANC02-Scramble and PANC02-gRNA2 cells using a H3K4me3-specific antibody. Based on our previous results, the H3K4me3 is mostly enriched at the downstream of the transcription start site of HLA-A, -B, and -C (Figure 2D) and H2K (Figure 6A). Accordingly, we designed PCR primers that covered the promoter regions approximately from −1000 to +3000 relative to the transcription start site of mouse H2K, as shown in Figure 6B. The ChIP analysis of mouse H2K promoter regions indicated that enrichment of H3K4me3 decreased in the promoter regions of H2K gene in the −1000 to 0 and +1000 to +3000 regions in PANC02-gRNA2 cells compared with PANC02-Scramble cells (Figure 6C). The decreased enrichment of H3K4me3 was especially significant in the −1000 to 0 and +1000 to +2000 regions relative to the transcription start site (Figure 6C).

### 2.7. WDR5 Deficiency Inhibits Immune Checkpoints and Immune Suppressive Cytokines’ Expression in Pancreatic Tumor Microenvironment

To explain why Wdr5 depletion did not effectively promote tumor growth in the syngeneic pancreatic tumor model, we further analyzed immune cell profiles in the tumor microenvironment. The analysis revealed that the expression of T-cell effectors, including FasL, GZMB, PRF1, and IFNγ, was remarkably repressed in the tumor microenvironment (Figure 7A). In addition, immune checkpoints PD-1, PD-L1, and OPN were downregulated dramatically in both total tumor (Figure 7A) and CD45^−^ cells (Figure 7B). Moreover, the expressions of immune-suppressive cytokines TGFβ and IL6 were also inhibited in total tumor (Figure 7A) and CD45^−^ cells (Figure 7B), especially IL6 (Figure 7A,B). We further analyzed human colon cancer (GSE178341 [39]) and human breast cancer (GSE176078 [40]) scRNA-Seq datasets to validate the correlation between Wdr5 and immune check points, as well as the correlation between Wdr5 and immune-suppressive cytokines. The analysis indicated that Wdr5 is positively correlated with PD-1, PD-L1, and Spp1 expression in both human colon cancer (Appendix A) and human breast cancer (Appendix A) in myeloid cells, epithelial cells, and T cells. A similar phenomenon can be observed in correlation between Wdr5 and both TGFβ and IL6 in both human colon cancer (Appendix A) and human breast cancer (Appendix A) in myeloid cells, cancer epithelial cells, and T cells. The above findings may contribute to improved CD8^+^/CD4^+^ T-cell infiltration and a subsequent reverse of the tumor-promoting effect mediated by Wdr5 deficiency-induced MHC I downregulation.

## 3. Discussion

Histone methyltransferases (HTMs) are critical enzymes that post-translationally methylate histones and are required for the epigenetic modulation of transcriptionally active or inhibitive forms of chromatin in eukaryotes [4,44]. MLL1 (Mixed-Lineage Leukemia 1) is an important member of the SET1 family of human histone methyltransferases (SET1A, SET1B, MLL1, MLL2, MLL3, and MLL4), which contribute to active chromatin via mono-, di-, and trimethylation of lysine 4 on histone H3 [6,45,46]. MLL1 has been most extensively studied because of its pivotal role in maintaining the expression of the Hox and Meis gene in normal hematopoiesis [47,48]. Dysregulation of MLL1 is associated with many human cancers, such as acute lymphoid leukemia (ALL) and acute myeloid leukemia (AML) [49]. However, MLL1 itself exhibits minimal or even no H3K4 methylation activity. To achieve maximal HTM activity, MLL1 is usually assembled into a core complex with a number of well-characterized regulatory subunits, including WDR5, Ash2L, RBBP5, and DPY30 [50,51]. These subunits are thought to constitute a common MLL/COMPASS subcomplex that forms a platform to mediate the Set1 enzyme and H3K4 substrate interaction [15]. In particular, the interaction between MLL1 and WDR5 is crucial for the integrity of the MLL1 complex and its HTM activity [52,53,54]. Therefore, identifying WDR5−MLL1 interaction antagonists such as WDR5-47 has been proposed as a potential therapeutic strategy for MLL-rearranged leukemias [55]. The growing experimental data indicate that WDR5 also might be a therapeutic target for lung adenocarcinoma [56], squamous cell carcinoma [57], and glioblastoma [58]. Our previous studies have shown that the expression of PD-L1 and OPN is regulated by the WDR5/MLL1-H3K4me3 pathway [18,19], and WDR5-47 effectively improved anti-PD-L1 efficacy in pancreatic tumor [18]. In this study, we report that WDR5 modulates the expression of MHC I and immune-suppressive cytokines in the pancreatic tumor microenvironment. This study went a step further to extend the regulatory roles of WDR5 in cancer-immunity cycles from leukemia to solid tumors, and it also expanded the tumor types potentially applied by WDR5-47.

In the tumor microenvironment, CTLs (Cytotoxic T Lymphocytes) kill tumor cells through recognizing and binding with MHC I, which represents tumor antigens and regulates tumor immunogenicity [59]. The loss of MHC I or decrease in immunogenicity is one of the hallmarks of human cancers [60,61]. Moreover, dysregulation of MHC I usually results in resistance to chemotherapy or immunotherapy [43,62]. Therefore, recovery of MHC I expression is a potential effective strategy to overcome tumor immune evasion. Epigenetic mechanisms are widely implicated in regulating the expression of MHC molecules in tumor cells. Recent data demonstrated that the presence of bivalent chromatin at promoters maintained by polycomb repressive complex 2 (PRC2) represses the MHC I antigen processing pathway, which inhibits the T cell-mediated antitumor immunity [24]. However, the PRC2 complex inhibitor EED226 enhances antitumor effects in nasopharyngeal carcinoma by upregulating MHC I gene expression [25]. Therefore, it is suggested that interventions facilitating more H3K4me3 depositions can improve MHC I expression. In this study, we determined that the ablation of WDR5 decreased MHC I expression both in vitro and in vivo through inhibition of H3K4me3 enrichment at the MHC I promoter region. WDR5 specific inhibitor WDR5-47 also suppressed MHC I expression in vitro. From this perspective, WDR5 shows an anti-cancer effect through positively modulating MHC I expression.

It is well known that transcription activation of immune checkpoints can be effectively induced by histone modification, and H3K4me3 enrichment contributes to PD-L1 and TOX2 expression in colorectal cancer [32,33]. In our previous study, we determined that OPN is a novel immune checkpoint which compensates for PD-L1 function to support tumor immune escape in the pancreatic tumor microenvironment [18], and the WDR5/MLL1-H3K4me3 epigenetic axis regulates both PD-L1 and OPN expression [18,19]. In this study, our data elucidated that WDR5 ablation led to a notable downregulation of both PD-L1 and OPN, thus further confirming our previous findings. On the other hand, the WDR5/MLL1-H3K4me3 axis regulates the expression of cytokines in the tumor microenvironment, such as TGFβ [63], TNFα1 [64], and IL6 [65]. Indeed, WDR5 depletion repressed the expression of TGFβ and IL6 in this study, both of which are immune-suppressive cytokines. From this perspective, WDR5 exhibits a pro-cancer effect by positively modulating the expression of immune checkpoints and immune-suppressive cytokines. It is worth noting that the pro-cancer effect is at least as potent as the anti-cancer effect in the pancreatic tumor microenvironment. Consequently, CD8+/CD4+ T-cell infiltration was significantly enhanced and neutralized the tumor-promoting effect mediated by Wdr5 deficiency-induced MHC I downregulation. This may explain why, in general, the tumor size/weight did not change significantly after WDR5 deficiency, and non-responders to chemotherapy/immunotherapy exhibited an even-higher WDR5 level compared with responders in colon-cancer and melanoma patients. More importantly, our results indicate that the WDR5-MHC I pathway and/or WDR5–immune checkpoint/cytokine pathways should be selectively targeted respectively in WDR5-based epigenetic cancer immunotherapy.

Strikingly, WDR5-47 therapy dramatically suppressed pancreatic tumor growth in vivo. This may be caused by the inhibition of both CD45^-^ and CD45^+^ cells in the tumor microenvironment. Similarly, a recent study identified a selective WDR5 degrader MS67, which effectively inhibited AML progression in patient-derived mouse models [66]. This may also be due to the removal of WDR5 from both CD45^−^ and CD45^+^ cells in the tumor microenvironment. That is why overall tumor inhibitive effects mediated by WDR5 inhibitor or degrader were different with those mediated by WDR5 deficiency only in CD45^-^ tumor cells in the tumor microenvironment. However, how WDR5-47 coordinates CD45^-^ and CD45^+^ cells to repress tumor growth in vivo requires further study. Nevertheless, in this research, we determined that the WDR5/MLL1-H3K4me3 epigenetic axis regulates pancreatic tumor tumorigenicity and immune suppression in the tumor microenvironment.

## 4. Materials and Methods

### 4.1. Patient Dataset Analysis

Hman colon cancer patient scRNA-Seq datasets (GEO accession: GSE178341 [39]), human breast cancer patient scRNA-Seq datasets (GEO accession: GSE176078 [40]), human melanoma patient scRNA-Seq datasets (GEO accession: GSE120575 [42]), and all correlations were extracted and analyzed from the Broad Institute Single Cell Portal.

### 4.2. Human Tumor Specimens

Human pancreatic carcinoma tumor specimens (n = 4) and non-neoplastic pancreatic tissues (n = 5) were provided by Tianjin Medical University Cancer Institute and Hospital. Studies with human specimens were approved by the Institutional Review Board of Tianjin Medical University Cancer Institute and Hospital (Approval #Ek2022223. Approval date: 7 March 2022) and The Institutional Review Board of Tianjin University (Approval# TJUE-2021-016. Approval date: 1 March 2021). The human specimens are de-identified samples.

### 4.3. Mice

Female C57BL/6 mice were purchased from Huafukang, Beijing, China. All mice used in this study were between 7 and 8 weeks old. All studies involving the use of mice were covered by a protocol approved by the Institutional Animal Care and Use Committees of Tianjin University (Approval #TJUE-2021-016. Approval date: 1 March 2021).

### 4.4. Cell Lines

The mouse pancreatic tumor PANC02 cell line was obtained from Jisidenuo, Shanghai, China, and is mycoplasma-free at the time of use.

### 4.5. Immunohistochemistry

Tissue sections were stained as previously described [19]. The sections were stained with anti-WDR5 antibody (sc-393080. Santa Cruz Biotechnology) and mounted in Vecta Mount Permanent Mounting Medium (Vector Lab, Burlingame, CA, USA). Instruments: Microtomes (Leica 2235, Wetzlar, Germany) and Autoclave (Triangle 18, Guangzhou, China).

### 4.6. CRISPR-Based Gene Knockout

psPAX2 (BR036, Fenghuishengwu, Changsha, China), pCMV-VSVG (BR081, Fenghuishengwu, Changsha, China), and lentiCRISPRv2 (Genscript, Piscataway, NJ, USA) plasmids containing Scramble (GAAGACTTAGTCGAATGAT), mouse WDR5-specific sgRNA1-coding sequence (AGGGAATATCTGATGTAGCG), or mouse WDR5-specific sgRNA2-coding sequence (ACTTGCCAACCATTCCCCAT) were co-transfected into HEK293T cells using Lipofectamine (Cat# T101-01, Vazyme, Nanjing, China). Cell-culture supernatants (virus particles) were collected and used to infect PANC02 cells. Stable cell lines were established using puromycin (Cat# BS111, Biosharp, Hefei, China) selection and then stained with anti-TriMethl-Histone H3-K4 mAb (Cat# A22146, Abclonal, MA, USA) and analyzed by Western blotting.

### 4.7. Western Blotting Analysis

Cells were lysed in NETN buffer, measured for protein concentration by Bradford Assay Kit (Cat# MA0079, Meilunbio, Dalian, China), and cell lysates were separated in 4-20% SDS–polyacrylamide gels (Bio-Rad, Hercules, CA, USA) and blotted to PVDF membranes (Bio-Rad). The antibodies are listed in Appendix A and original Western blots are shown in Appendix A.

### 4.8. Gene Expression Analysis

qPCR analysis was performed as previously described [18]. The sequences of primers are listed in Appendix A.

### 4.9. Cell Viability Assay

Cell viability assays were performed using Cell Counting Kit-8 (Cat#GK10001, GLPBIO, Shanghai, China) according to the manufacturer’s instructions.

### 4.10. Flow Cytometry

For cell lines, cells were first harvested, centrifuged, and washed in PBS. The cell pellets were resuspended in 100 µL PBS and stained with fluorescent dye-conjugated antibodies. The suspension was then washed by PBS and resuspended in 300 µL PBS. For tumor, tissues were collected and digested in digestion buffer (collagenase at 1 mg/mL, BS163, Biosharp, Hefei, China; hyaluronidase at 0.1 mg/mL, BS171, Biosharp, Hefei, China; and DNase I at 30 U/mL, D8071, Solabrio, Beijing, China) and passed through a 100 µm cell filter. Cells were then lysed with red cell lysis buffer and stained with fluorescent dye-conjugated antibodies. All antibodies were obtained from Biolegend and are listed in Appendix A. Stained cells were analyzed by flow cytometry (BD FACS Verse). All flow cytometry data were analyzed using FlowJo v10 software (BD Biosciences, San Diego, CA, USA).

### 4.11. In Vivo Tumor Mouse Model

To establish a tumor-bearing mouse model, PANC02-Scramble and PANC02-WDR5 gRNA2 cells (1.5 × 10^7^ cells/mouse) were subcutaneously injected to the right flank of C57BL6 mice. In addition, PANC02 WT cells (1×10^6^ cells/mouse) were subcutaneously injected into the right flank of C57BL6 mice and treated with solvent and WDR5-47 (20 mg/kg body weight) by i.p. injection 17 days after tumor cell injection daily, totaling 9 times.

### 4.12. CD45^+^ Cells Isolation

Total tumor tissues were digested in a tissue digestion buffer at room temperature for approximately 40 min, with agitation. The digests were then grinded and passed through a 70 μm cell strainer (251200, Sorfa, Huzhou, China). The cells were lysed with red cell lysis buffer for 5 min in room temperature and resuspended in PBS with 1% BSA. The CD45^+^ immune cells were then purified from the tumor cell suspension using a Mouse CD45 Selection Kit (Cat# 480028, BioLegend, San Diego, CA, USA) and separated by a magnetic stand.

### 4.13. WDR5-47 Synthesis

WDR5-47 was synthetized according to the literature. Briefly, 2-fluoro-5-nitroniline was reacted with N-methyl piperazine to generate the corresponding amino intermediate, which was then reacted with 2-chloro-4-fluoro-3-methylbenzoyl chloride to the yield of the target compound, WDR5-47 [17].

### 4.14. Chromatin Immunoprecipitation (ChIP) Assay

ChIP assays were carried out based on Chromatin Immunoprecipitation Assay Kit (Cat# 17-295, Merck, NJ, USA). In summary, cells (1 × 10^6^ cells) were harvested, crosslinked by 1% formaldehyde (Cat# F809702, Macklin, Shanghai, China), washed in PBS, resuspended in SDS Lysis Buffer and sonicated, and then centrifuged. The sonicated cell supernatant was diluted tenfold in ChIP Dilution Buffer (reserve 5% diluent as input). We then added the immunoprecipitating antibody and incubated at 4 °C overnight. The beads were then washed in Low-Salt Immune Complex Wash Buffer, High-Salt Immune Complex Wash Buffer, LiCl Immune Complex Wash Buffer, and TE Buffer, sequentially, and eluted in 500 µL Elution Buffer. The eluants were reversed, and we recovered DNA by phenol/chloroform/isoamyl alcohol (Cat# P59330, Acmec, Shanghai, China). The H2K promoter DNA was detected by Real-Time Quantitative PCR, using the promoter DNA-specific primers listed in Appendix A.

### 4.15. Statistical Analysis

Statistical comparisons in Figure 1C and Appendix A were carried out by the two-sided Wilcoxon rank-sum test. All boxplots showed the lower, median, and upper quartiles of the values. Error bars on the bar plots represent standard deviation of the mean. Statistical comparisons in Figure 2C were carried out by the Spearman’s rank Correlation Test. All statistical tests were implemented with R statistical programming language (R version 4.2.0). Statistical analysis for other figures were performed by two-sided Student’s *t*-test, using the GraphPad Prism program 5.0 (GraphPad Software, Inc., San Diago, CA, USA). A *p* < 0.05 is considered statistically significant. The chart of experimental methods for the whole manuscript is shown in Appendix A.

## 5. Conclusions

It is well reported that the WDR5-H3K4me3 epigenetic axis has various regulatory functions in the cancer-immunity cycle and is often dysregulated in the tumor microenvironment. Although the function of the WDR5/MLL1-H3K4me3 axis has been extended from hematopoietic cancer to solid tumor, its roles in tumor cells and immune cells, respectively, in solid tumor is still largely unknown. In this study, we first demonstrated that WDR5 is present at a higher level in human pancreatic tumor tissues compared with adjacent normal pancreases. Moreover, WDR5 expression is negatively correlated with patients’ response to chemotherapy or immunotherapy in human colon cancer and melanoma. Nevertheless, WDR5 expression exhibits a positive correlation with HLA level in human cancer cells, and H3K4me3 enrichment is found at the promoter regions of the HLA-A, HLA-B, and HLA-C genes in pancreatic cancer cells based on an Encode database analysis. Furthermore, WDR5 deficiency or inhibition causes the significant repression of MHC I expression in vitro and in vivo in pancreatic tumor cells. At the molecular level, we revealed that WDR5 deficiency suppresses H3K4me3 deposition at the MHC I promoter region to inhibit MHC I transcription. In addition, WDR5 ablation effectively results in the downregulation of immune checkpoints (PD-1, PD-L1, and Spp1) and immunosuppressive cytokines (TGFβ and IL6) in pancreatic tumor microenvironments, leading to improved CD8^+^/CD4^+^ T-cell infiltration. The scRNA Seq dataset analysis also confirmed the positive correlation between Wdr5 and PD-1, PD-L1, and Spp1, as well as the positive correlation between Wdr5 and both TGFβ and IL6 in both human colon cancer and human breast cancer. Taken together, our findings thus determined that WDR5 not only regulates tumor-cell immunogenicity to inhibit tumor growth but also stimulates immune suppressive pathways to promote tumor immune escape from host immune surveillance, providing the rationale for the selective targeting WDR5-MHC I pathway and/or WDR5-immune checkpoint/cytokine pathways, respectively, in WDR5-based epigenetic tumor immunotherapy.

## Figures and Tables

**Figure 1 ijms-25-08773-f001:**
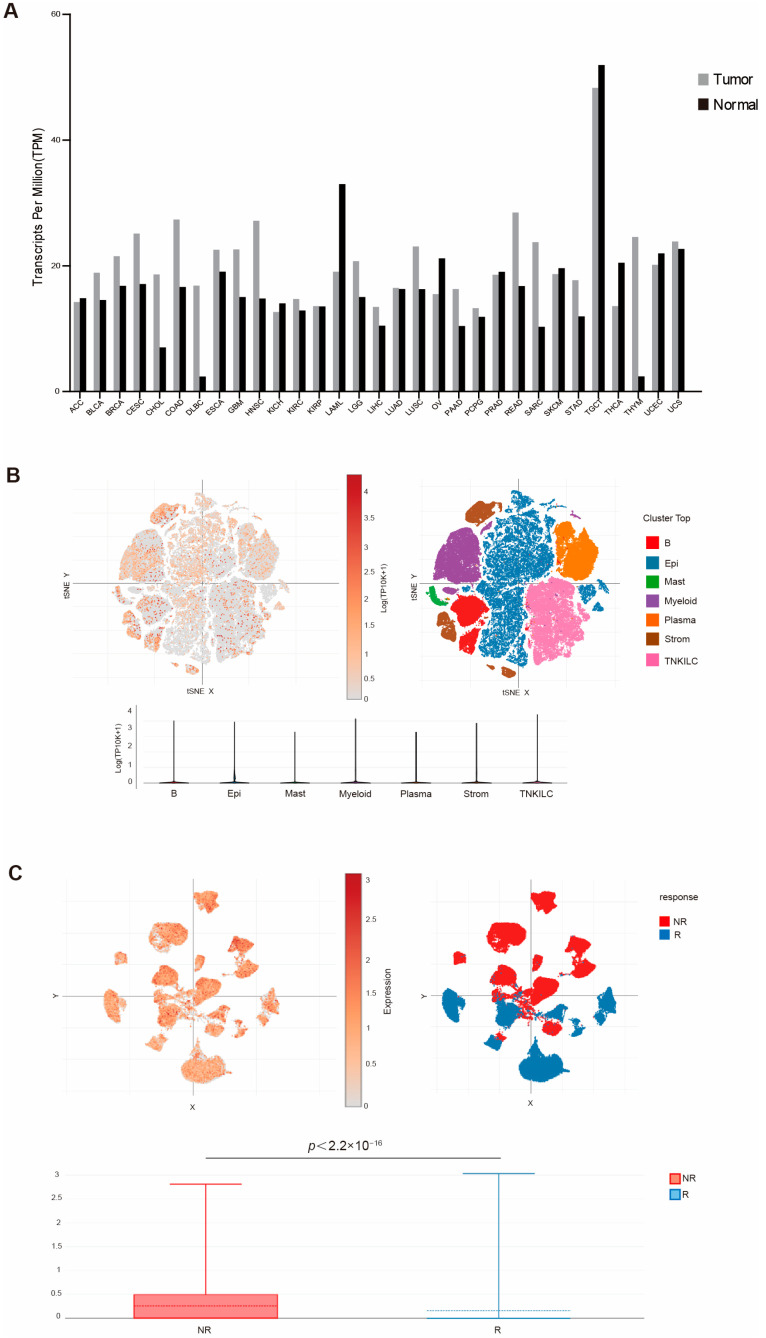
WDR5 expression profiles in human cancer. (**A**) WDR5 expression level in the indicated tumor tissues and normal tissues. (**B**) UMAP of major cell subpopulations (upper-right panel) and WDR5 expression level (upper-left panel) in the indicated cell subpopulations in human colon cancer. The bottom panel shows violin plot of WDR5 expression level in the indicated major cell subpopulations as in the upper-left panel. (**C**) UMAP of cell subpopulations showing WDR5 expression level in responders and non-responders to combined PD-1, BRAF, and MEK inhibition in human colorectal cancer patients.

**Figure 2 ijms-25-08773-f002:**
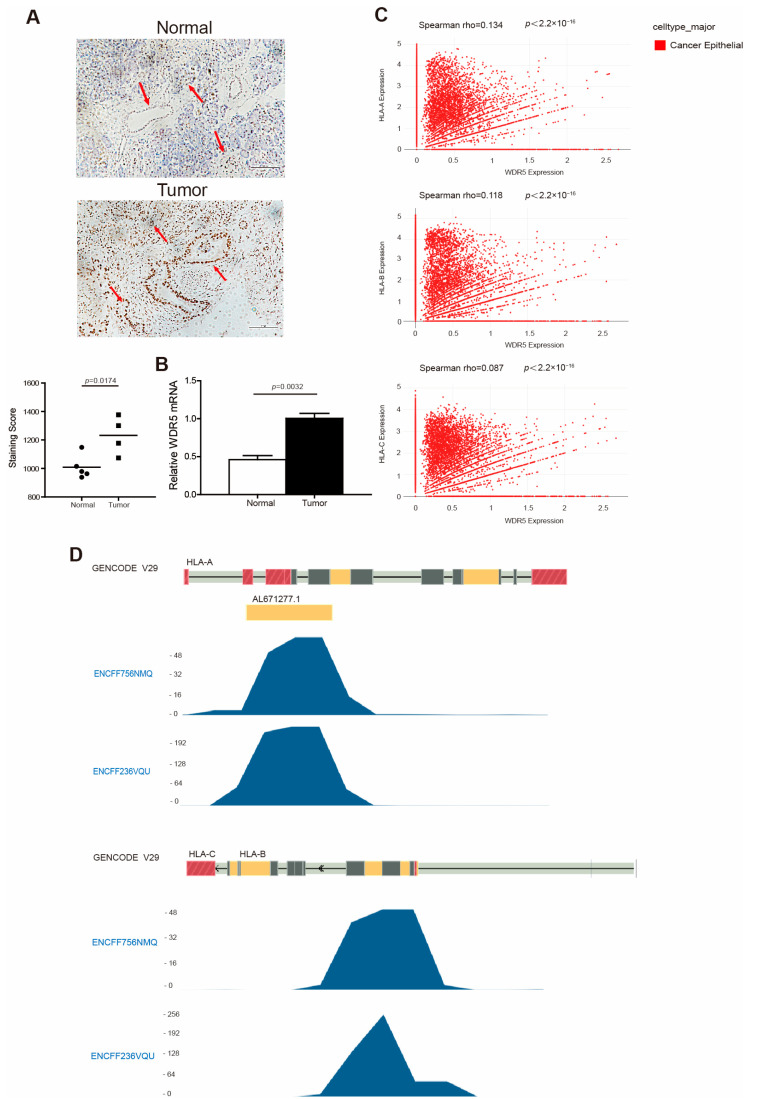
WDR5 has higher expression levels in human tumor tissues and is positively correlated with HLA expression in human cancer. (**A**) Human non-neoplastic pancreas (n = 5) and pancreatic tumors (n = 4) were stained with WDR5-specific antibody (top panel). WDR5 staining intensity is indicated by brown color. Shown in upper panel is one pair of representative images. The bottom panel shows the quantification of WDR5^+^ cells in the adjacent normal pancreas and pancreatic tumor, as shown in the top panel. Staining score: Wdr5^+^ cells in each microscopic field of view. (**B**) qPCR analysis of WDR5 expression in human pancreatic tumors and adjacent normal pancreas. (**C**) Correlation between expression of WDR5 and three HLA subtypes in cancer epithelial cells in human breast cancer. (**D**) H3K4me3 deposition profiles in the HLA promoter region in human pancreatic tumor cell line PANC1. GENCODE V29 indicates GENCODE-Human Release 29 (https://www.gencodegenes.org/human/release_29.html, accessed on 15 March 2024). ENCFF756NMQ and ENCFF236VQU indicate different ChIP-Seq data.

**Figure 3 ijms-25-08773-f003:**
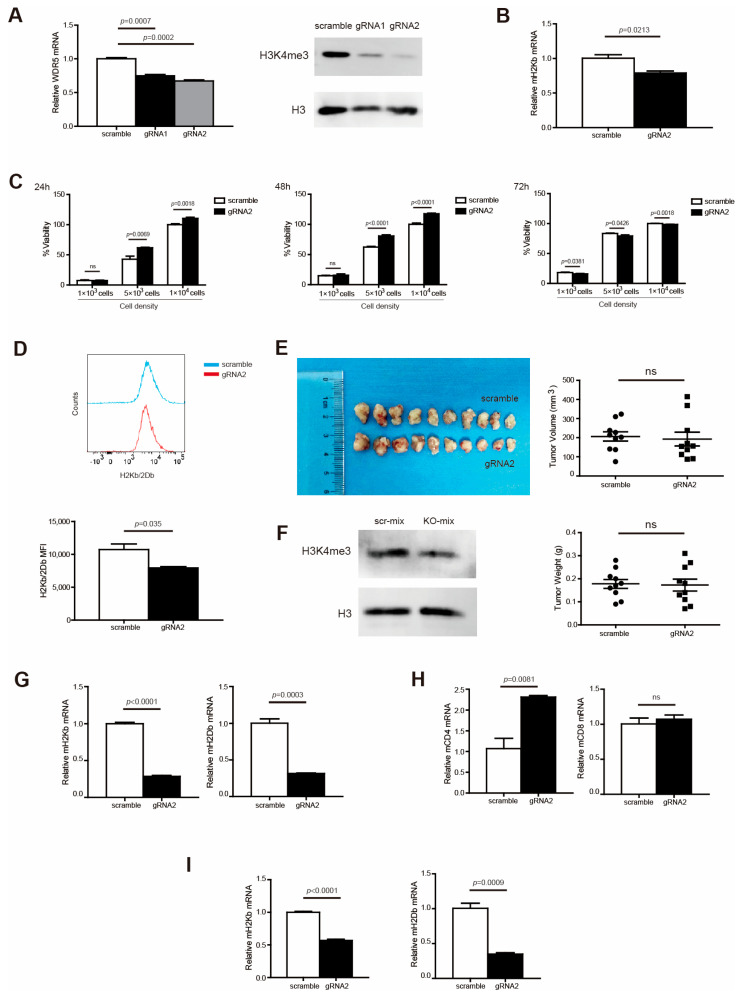
Wdr5 increases MHC I expression in murine tumor cells vitro and in vivo. (**A**). Total RNA was prepared from PANC02-Scramble, PANC02-gRNA1, and PANC02-gRNA2 cells and analyzed for WDR5 RNA level by qPCR, with β-actin as internal control (left panel). PANC02-Scramble, PANC02-gRNA1, and PANC02-gRNA2 cells were cultured in vitro for 24 h. Cells were analyzed by Western blotting for H3K4me3 level (right panel). (**B**). Total RNA was prepared from PANC02-Scramble and PANC02-gRNA2 cells and analyzed for H2Kb RNA level by qPCR, with β-actin as internal control. (**C**) PANC02-Scramble and PANC02-WDR5 KO tumor cell lines, as indicated, were cultured in vitro for 24–72 h and measured for cell viability. (**D**) PANC02-Scramble and PANC02-gRNA2 cells were cultured in vitro for 24 h. Cells were stained with anti-H2Kb/2Db mAb and analyzed by flow cytometry. Shown are representative flow cytometry plots (top panel) and the H2Kb/2Db MFI (bottom panel). (**E**) PANC02-Scramble and PANC02-gRNA2 cells (1.5 × 10^7^ cells/mouse) were injected to mice subcutaneously to establish tumor. Shown are tumor images (top left panel). The tumor volume and weight were quantified and are shown in the right panel. (**F**) Total tumors from (**E**) were digested with collagenase and analyzed by Western blotting for H3K4me3 level. (**G**–**I**) Total RNA was prepared from PANC02-Scramble and PANC02-gRNA2 tumors, as shown in (**E**), and analyzed for H2Kb; H2Db expression was determined by qPCR, with β-actin as internal control (**G**). Immune cells (CD45^+^) (**H**) and tumor cells (CD45^−^) (**I**) were isolated from the tumor tissues using CD45 mAb-conjugated magnetic beads and analyzed for the indicated gene expressions by qPCR.

**Figure 4 ijms-25-08773-f004:**
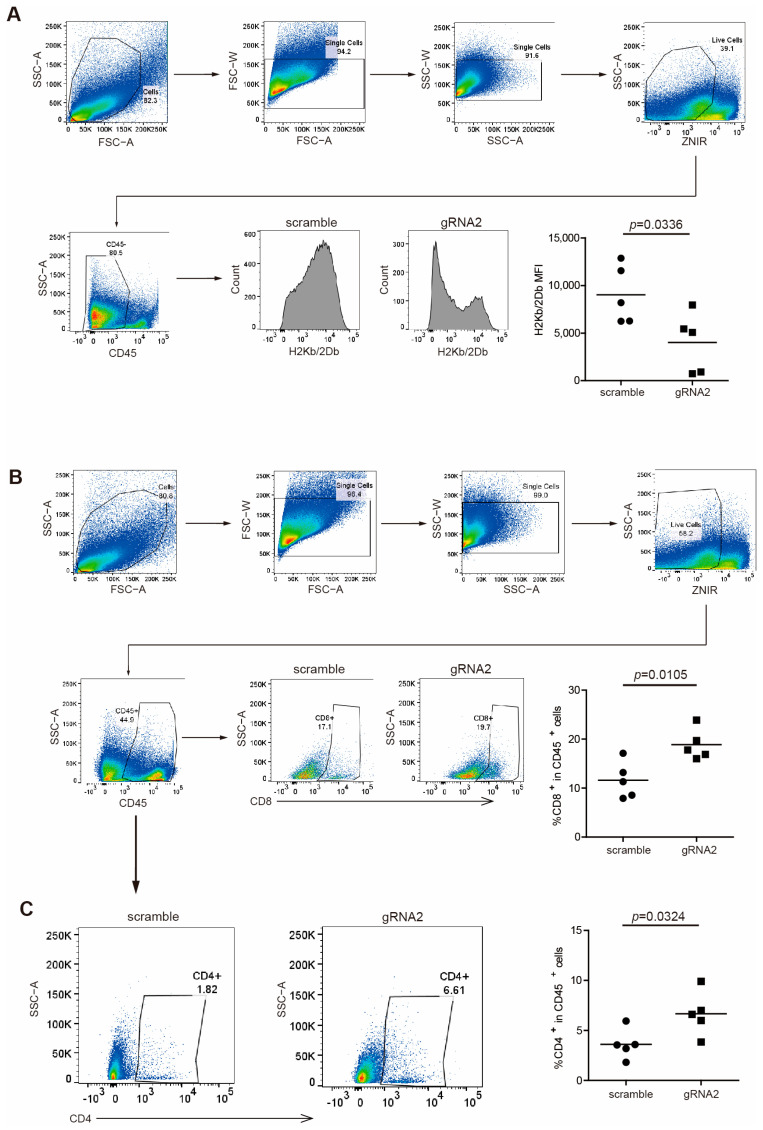
Wdr5 increases MHC I expression and T-cell infiltration in mouse tumor vivo. (**A**) PANC02-Scramble and PANC02-gRNA2 tumors as shown in Figure 3E were collected, digested with collagenase, stained with anti-H2Kb/2Db mAb, and analyzed by flow cytometry. Shown is the gating strategy (upper panel) and H2Kb/2Db MFI (bottom panel). (**B**) PANC02-Scramble and PANC02-gRNA2 tumors were processed as in (**A**), stained with anti-CD8 mAb, and analyzed by flow cytometry. Shown is the gating strategy (upper panel) and percentage of CD8^+^ cells (bottom panel). (**C**) PANC02-Scramble and PANC02-gRNA2 tumors were processed as in (**A**), stained with anti-CD4 mAb, and analyzed by flow cytometry. Shown is percentage of CD4^+^ cells.

**Figure 5 ijms-25-08773-f005:**
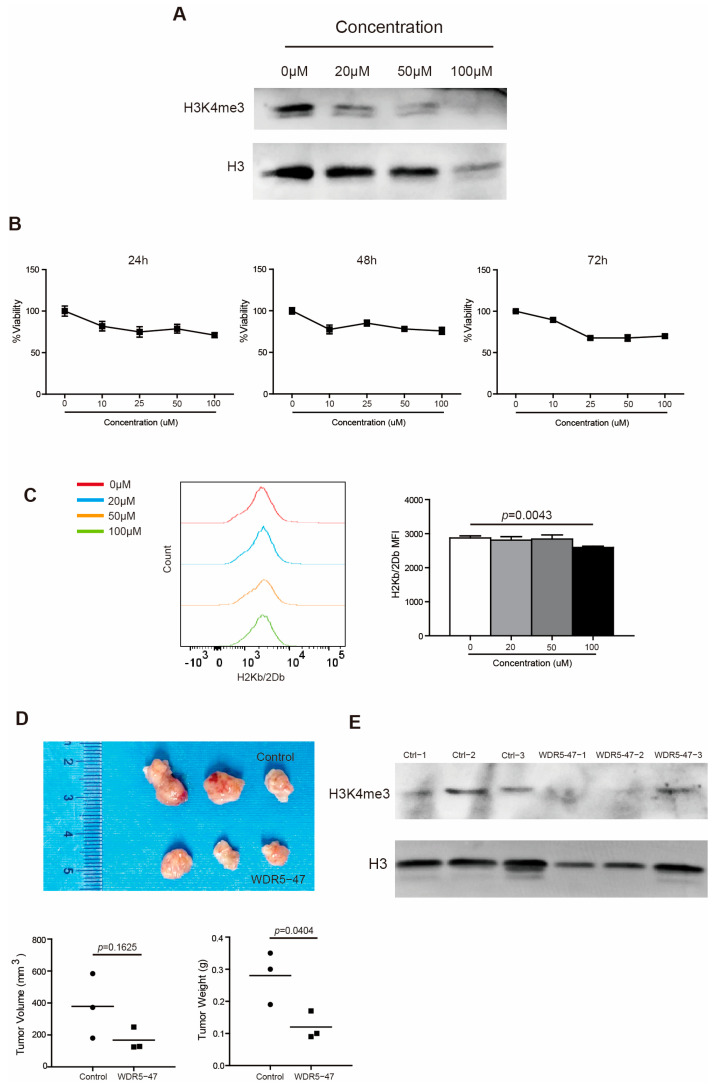
Wdr5 inhibition decreases MHC I expression. (**A**) PANC02 cells were cultured in the presence of WDR5-47, at the indicated concentrations for 72 h, and analyzed by Western blotting for H3K4me3. (**B**) PANC02 cells were cultured in the presence of WDR5-47, at the indicated concentrations, for 24–72 h and analyzed for cell viability. (**C**) PANC02 cells were cultured in the presence of WDR5-47, at the indicated concentrations, for 24 h and analyzed for their H2Kb/2Db expression by flow cytometry. (**D**) PANC02 cells (1 × 10^6^ cells/mouse) were injected into mice subcutaneously. The tumor-bearing mice were treated 17 days later with solvent, and WDR5-47 (20 mg/kg body weight) daily for a total of 9 times by i.p. injection. Mice were sacrificed on day 27. Shown are tumor images (top panel). The tumor volume and weight were quantified and are shown in the bottom panel. (**E**) Tumors, as shown in (**D**), were digested with collagenase and analyzed by Western blotting for H3K4me3.

**Figure 6 ijms-25-08773-f006:**
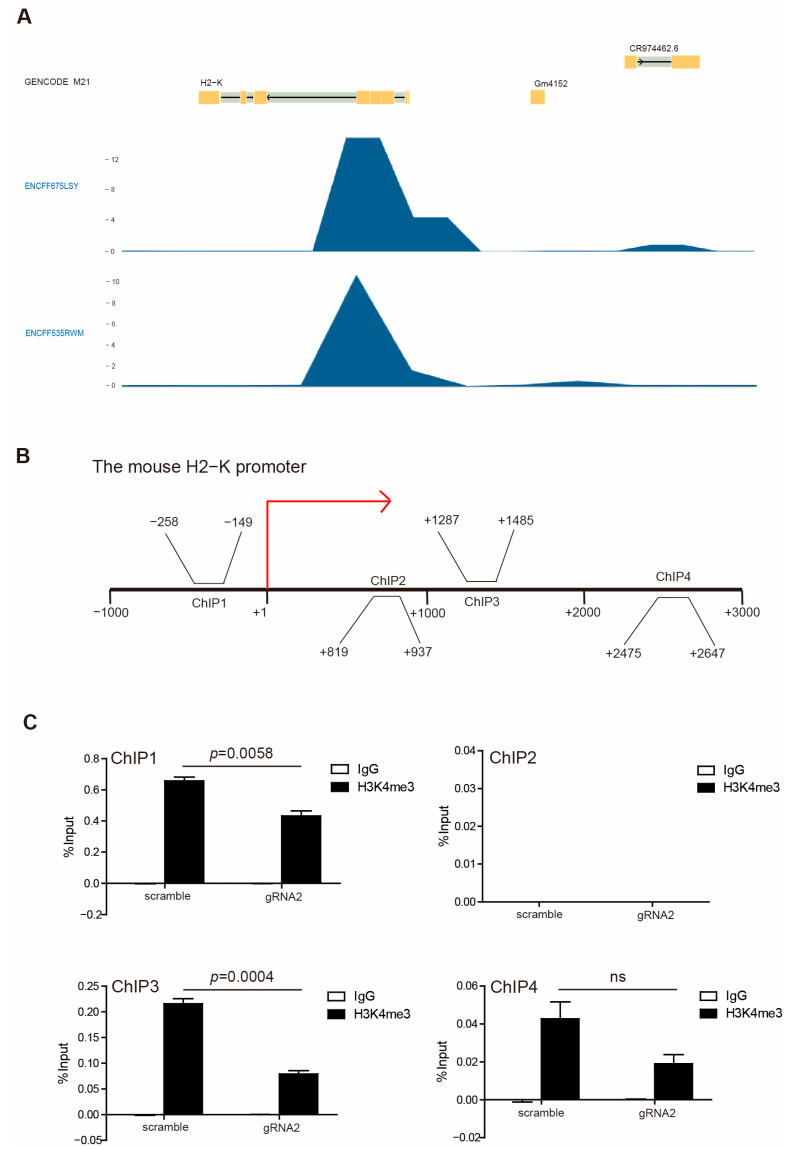
H3K4me3 binds to H2K1 promoter. (**A**) H3K4me3 deposition profiles in the H2K promoter region in mouse melanoma MEL cells. GENCODE M21 indicates GENCODE—Mouse Release M21 (https://www.gencodegenes.org). ENCFF675LSY and ENCFF535RWM indicate different ChIP-Seq data. (**B**) Shown are four pairs of primers, spanning from −3000 to +1000 relative to the mouse H2-K promoter region. (**C**) PANC02-Scramble and PANC02-gRNA2 were cultured for 24 h. Chromatin was then prepared, and ChIP was performed with a H3K4me3-specific antibody. The immunoprecipitated chromatin fragments were then analyzed by qPCR, with primers shown in (**B**).

**Figure 7 ijms-25-08773-f007:**
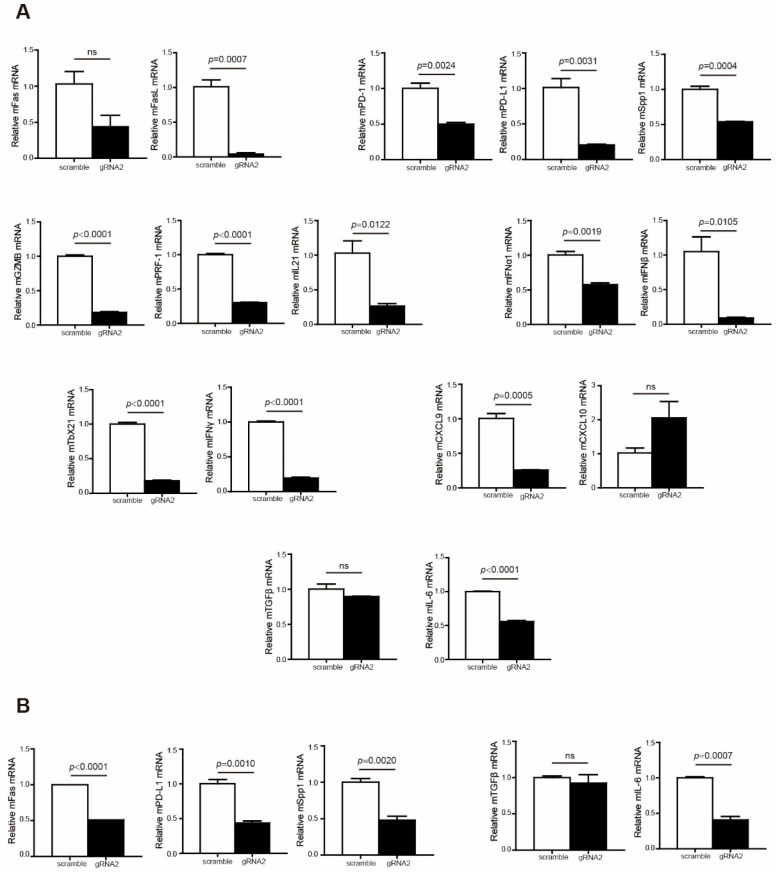
Wdr5 deficiency leads to a less-immune-suppressive tumor microenvironment. (**A**). Total RNA was prepared from PANC02-Scramble and PANC02-WDR5 KO tumors in Figure 3E and analyzed for RNA level of the indicated genes by qPCR, with β-actin as the internal control. (**B**). Tumor cells (CD45^−^) were isolated from the total tumor tissues using CD45 mAb-conjugated magnetic beads and analyzed for the indicated genes by qPCR.

## Data Availability

Publicly available datasets were analyzed in this study. The datasets are described in the Section 4.

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
