# Peer review of "The Wdr5-H3K4me3 Epigenetic Axis Regulates Pancreatic Tumor Immunogenicity and Immune Suppression"

_ijms, 2024, doi:10.3390/ijms25168773_

Round 1

Reviewer 1 Report

Comments and Suggestions for Authors

In the current study, Deng et al. suggested that WDR5/MLL1-H3K4me3 epigenetic axis has important role in PDAC tumorigenecity and immune response.

Although the current study is interesting, there is NO indication of statistical significance in almost graphs. So, there is no basis for judging the importance of research.

The most important thing is the current study didn't include any approved information of IACUC for animal study and IRB for human study both.

Overall, the current manuscript did not reach the level of review.

Reviewer 2 Report

Comments and Suggestions for Authors

The paper is well written and also the references cited are current and appropriate. However the material and methods part could be amplied

Human Tumor Specimens-->How many?

IHC-->Wich instrument was used?

Reviewer 3 Report

Comments and Suggestions for Authors

In this article, the role of WDR5 in human pancreatic tumor tissues has been investigated, and the findings are significant. WDR5 deficiency or inhibition was found to repress MHC I expression both in vitro and in vivo in pancreatic tumor cells. This regulation of tumor cell immunogenicity by WDR5 was found to suppress tumor growth and activate immune suppressive pathways, leading to tumor immune evasion. These findings have important implications for pancreatic tumor research, furthering our understanding of the disease and potentially opening new avenues for therapeutic interventions; however, there is a concern. They should add the chart of experimental methods. No other comments are available. 

Comments on the Quality of English Language

Minor English editing is required. 

Round 2

Reviewer 1 Report

Comments and Suggestions for Authors

Major concerns;

1. Line 195-197

There is no reason why authors focused on only COAD and PAAD. In the figure 1A, many type of tumors were upregulated WDR5 expression compared to normal control. Therefore, author should address why only COAD and PAAD were selected and analyzed.

2. L204-207

Is it significant change WDR5 expression in non-responders as compared to responders in some treatment option? As shown in the Figure 1C, it seems not significant between two groups. If there is not significant, this result is of littel value for Figure 1. It is also necessary to subtract the results and reconstruct the Figure 1.

3. L222-225

The image quality is poor (Fig. 2A). So, it should be changed clear one. In addition, the counted WDR5+ cells nees to change another category using staining intensity or staining score etc.

4. L226-229

In the Figure 2C, the spearman rho values were less then 0.14. It means all correalation seems to be not significant. Neverthess, in order to these results to be meaningful, correlation must be statistical significnat, so p-value should be marked. 

5. L231-236 & Figure legend 2D.

What is the Encode dataset? There is no information. In addition, there is no information about Figure 2D. What's mean GENCODE V29? ENCFF756NMQ? Current results that do not provied any basic information are compeletly incomprehensible and uninterpretable.

There are still many areas of concern that are difficult to list one by one.

Overall, the current manuscript has still not reach the level of review.
